# Magnesium: Health Effects, Deficiency Burden, and Future Public Health Directions

**DOI:** 10.3390/nu17223626

**Published:** 2025-11-20

**Authors:** Marijana Matek Sarić, Tamara Sorić, Željka Juko Kasap, Nataša Lisica Šikić, Mladen Mavar, Jurgita Andruškienė, Ana Sarić

**Affiliations:** 1Department of Health Studies, University of Zadar, Splitska 1, 23000 Zadar, Croatia; marsaric@unizd.hr (M.M.S.); nlisicasi@unizd.hr (N.L.Š.); 2Psychiatric Hospital Ugljan, Otočkih Dragovoljaca 42, 23275 Ugljan, Croatia; ravnatelj@pbu.hr; 3Clinic DIVA Ltd., Postrojbi Specijalne Policije Zadar 11, 23000 Zadar, Croatia; zeljkajuko@yahoo.com; 4Department of Public Health, Faculty of Health Sciences, Klaipeda University, Herkaus Manto str. 84, 92294 Klaipeda, Lithuania; jurgita.andruskiene@ku.lt; 5School of Medicine, Catholic University of Croatia, Ilica 242, 10000 Zagreb, Croatia; asaric1@unicath.hr

**Keywords:** magnesium, magnesium deficiency, magnesium supplementation, metabolism, cardiometabolic health, bone health, neurological function, public health, nutrition, chronic disease prevention

## Abstract

Magnesium (Mg^2+^) is the fourth most abundant cation in the human body and a critical cofactor in hundreds of enzymatic reactions that regulate energy metabolism, neuromuscular function, cardiovascular health, bone integrity, immune defense, and psychological well-being. Despite its essential roles, magnesium deficiency remains common worldwide, driven by inadequate dietary intake, chronic diseases, medication use, and lifestyle factors. Low magnesium status is associated with hypertension, type 2 diabetes, osteoporosis, migraines, depression, and chronic inflammation, whereas sufficient intake supports cardiometabolic resilience, skeletal strength, neurological stability, and healthy aging. This review synthesizes current evidence on magnesium metabolism, physiological functions, and the health consequences of deficiency, and it summarizes global status with attention to biomarker limitations, widespread suboptimal intake, and key demographic and lifestyle determinants. It also discusses dietary sources, supplementation, and innovative approaches such as food fortification, personalized nutrition, and improved diagnostic strategies. The evidence highlights magnesium as a modifiable factor with potential to lessen the burden of chronic diseases. Recognizing magnesium deficiency as a pressing but underappreciated public health issue, this article underscores the need for integrated strategies to optimize magnesium balance at both individual and population levels.

## 1. Introduction

Magnesium (Mg^2+^) is a divalent metal ion and the fourth most abundant cation in the human body, after calcium, potassium, and sodium [1]. The body contains approximately 25 g of magnesium, with 50–60% stored in bone and the remainder distributed in soft tissues [1]. As an essential mineral, magnesium acts as a cofactor in more than 600 enzymatic reactions fundamental to energy metabolism, muscle contraction, and nerve transmission [2,3]. Through these processes, it plays a pivotal role in regulating blood pressure, supporting cardiovascular function, modulating glucose control in diabetes, strengthening bone, preventing migraines, and maintaining oral health [4].

One of magnesium’s most critical functions is its involvement in the production of adenosine triphosphate (ATP), the body’s principal energy carrier [5]. By stabilizing ATP, magnesium enables efficient energy transfer, which is fundamental for cellular metabolism and neuromuscular activity [6]. Adequate magnesium status supports muscle contraction and relaxation, regulates electrolyte balance, and enhances exercise capacity as well as post-exercise recovery, whereas deficiency is associated with fatigue, weakness, and impaired physical performance [5,7]. Beyond this, magnesium also contributes to protein synthesis, bone mineralization, and psychological well-being, underscoring its importance across both physical and mental domains of health [5].

Because of its broad physiological significance, magnesium deficiency apparently is highly prevalent, largely due to modern dietary patterns, chronic stress, medication use, and certain health conditions [8]. Growing evidence links insufficient intake to hypertension, type 2 diabetes, osteoporosis, migraines, depression, and chronic inflammation [4,8]. However, uncertainties remain regarding optimal intake thresholds, the sensitivity of current diagnostic methods, and the most effective strategies for prevention and intervention, questions that continue to generate debate within the field.

This review aims to provide a comprehensive synthesis of current knowledge on magnesium metabolism and homeostasis, emphasizing its systemic health effects, the consequences of deficiency, and the global prevalence of low magnesium status. Furthermore, it examines dietary sources, supplementation, and emerging approaches such as food fortification, personalized nutrition, and novel diagnostics. By situating magnesium deficiency within a broader public health context, this review highlights the need for integrated strategies to optimize magnesium balance and reduce the global burden of chronic diseases.

## 2. Magnesium Metabolism and Homeostasis

Magnesium metabolism is fundamental to diverse cellular and systemic processes. As the second most abundant intracellular cation, magnesium is involved in carbohydrate, lipid, protein, and nucleic acid metabolism [9,10,11,12] and supports intracellular energy transfer via ATP-dependent reactions [4,13].

Beyond bioenergetics, magnesium also contributes to the regulation of neuromuscular transmission, cardiovascular function, immune activity, and ion homeostasis [4,9,13,14,15]. In particular, it modulates calcium-dependent signaling and stabilizes cellular membranes, thereby supporting normal myocardial contractility and electrical conduction [14,16].

Systemic homeostasis is maintained through coordinated regulation of intestinal absorption, renal handling, and skeletal storage [6,17,18]. Intestinal uptake occurs via passive paracellular diffusion and active transcellular transport, and is modulated by dietary composition: phytates, oxalates, high-fiber foods, and high doses of calcium, phosphorus, and iron can inhibit absorption, whereas lactose, certain carbohydrates, like oligosaccharides and inulin, proteins, and medium-chain triglycerides can enhance it [1,19,20,21,22].

The kidneys are the principal regulators of magnesium balance. Approximately 80% of plasma magnesium is filtered at the glomerulus, with most reabsorbed along the nephron [23,24]. The thick ascending limb of the loop of Henle reabsorbs 50–70% of filtered magnesium via a claudin-16–dependent paracellular pathway [25,26,27,28]. Final adjustments occur in the distal convoluted tubule, where 5–10% is reabsorbed through active transport mediated by the TRPM6 channel [18,29]. Although quantitatively smaller, this site critically determines urinary magnesium excretion and overall systemic balance [17,24,25]. Renal handling is further influenced by hormonal and metabolic factors. Parathyroid hormone, calcitonin, and epidermal growth factor stimulate reabsorption, whereas metabolic acidosis or potassium depletion reduce it [23,30].

In addition to renal and intestinal regulation, bone serves as a dynamic reservoir, continuously exchanging magnesium with plasma to stabilize extracellular concentrations [18,31]. This integrated system ensures stable serum levels under varying dietary and physiological conditions. Disruption at any of these regulatory sites, whether through reduced intestinal absorption, increased renal losses, or impaired skeletal buffering, may therefore result in systemic deficiency even when serum concentrations remain within normal range, which may partly explain inconsistencies observed across clinical studies.

## 3. Magnesium and Its Role in Physiological and Mental Well-Being

### 3.1. Blood Pressure and Cardiovascular Health

Magnesium plays an important role in cardiovascular regulation, primarily through its vasodilatory properties and its function as a natural calcium channel blocker [4,32]. Consistent with these mechanisms, magnesium deficiency has been associated with an increased risk of cardiovascular diseases, including stroke and coronary artery disease [33,34]. Observational studies consistently report inverse associations between magnesium intake and cardiovascular events [34,35], but these findings should be interpreted with caution because observational designs are susceptible to residual confounding from factors such as diet quality, lifestyle behaviors, and comorbidities that may not be fully controlled.

Evidence from randomized controlled trials (RCTs) and meta-analyses generally support these findings, but results vary considerably across studies. A meta-analysis of 34 RCTs including 2028 participants found that magnesium supplementation, at a median dose of 368 mg/day over three months, led to significant reductions in systolic (−2.00 mmHg) and diastolic (−1.78 mmHg) blood pressure [36]. Similarly, an earlier review concluded that daily supplementation with 500–1000 mg of magnesium may lower systolic blood pressure by 2.7–5.6 mmHg and diastolic blood pressure by 1.7–3.4 mmHg [32]. However, not all trials have confirmed these effects. A 24-week RCT in overweight and obese adults reported no significant effect of 350 mg/day magnesium supplementation on blood pressure, though arterial stiffness was significantly reduced [37]. Likewise, another RCT involving 164 overweight or obese participants found no significant differences in blood pressure or arterial stiffness across magnesium citrate, oxide, and sulfate supplementation groups [38]. These results should be interpreted carefully, as the trials differ substantially in methodological quality, baseline magnesium status, dose and chemical form, intervention duration, and participant phenotype (e.g., normotensive vs. hypertensive), with effects tending to be greater in deficient or higher-risk groups and smaller in magnesium-replete populations. Such heterogeneity limits direct comparability and reduces the certainty of the overall effect.

Magnesium is also essential for maintaining normal cardiac rhythm. Hypomagnesemia can depolarize cardiac cells, increasing the risk of arrhythmias such as supraventricular and ventricular tachyarrhythmias, whereas hypermagnesemia can hyperpolarize cardiac cells and suppress arrhythmogenic activity [39]. Similarly to findings in blood pressure research, RCT results on arrhythmia management are inconsistent. While intravenous administration of magnesium sulfate in patients with acute myocardial infarction showed no effects on arrhythmia [40], in another RCT conducted on patients who experienced heart failure it led to a significant reduction in episodes of ventricular tachycardia and ectopic beats [41]. Importantly, the evidence in this area is largely derived from acute or inpatient settings using intravenous magnesium, which limits extrapolation to long-term oral supplementation or preventive strategies in community populations.

Overall, current evidence supports the role of magnesium in multiple cardiovascular pathways, including vascular tone and cardiac rhythm stability, but the strength and consistency of findings vary depending on study design, population characteristics, and baseline magnesium status. Continued large-scale, rigorously designed RCTs with standardized dosing, appropriate phenotype stratification, robust assessment of magnesium status, and sufficient duration are needed to more clearly establish magnesium’s preventive and therapeutic relevance in cardiovascular health.

### 3.2. Glucose Metabolism and Type 2 Diabetes

Magnesium deficiency is increasingly recognized as a contributing factor in the development of insulin resistance and disturbances in glucose metabolism, thereby raising the risk of type 2 diabetes [4,42]. Adequate magnesium levels are essential for glycaemic control and metabolic health, participating in enzymatic processes involved in glucose transport, phosphorylation, and insulin receptor signaling [43,44].

Mechanistic studies show that magnesium deficiency disrupts ATP-sensitive potassium channel activity in pancreatic β-cells, leading to impaired insulin secretion and reduced β-cell function [45]. This is particularly relevant in individuals with diabetes, who often present with reduced serum magnesium concentrations. Hypomagnesemia has been reported in 14–48% of patients with type 2 diabetes, compared with about 2.5–15% in the general population, and may further aggravate hyperglycemia and insulin resistance [46]. Among intensive care patients, prevalence can reach 65%, underscoring its clinical importance [43]. However, this prevalence estimates derive from heterogeneous settings and diagnostic criteria for hypomagnesemia, which complicates direct comparison across populations.

Epidemiological evidence consistently demonstrates an inverse relationship between magnesium intake and type 2 diabetes risk. A meta-analysis of 13 prospective cohort studies (*n* = 536,318) found that each 100 mg/day increase in magnesium intake was associated with a relative risk of 0.86 for type 2 diabetes (95% confidence interval: 0.82–0.89) [47]. Nevertheless, observational findings may be influenced by residual confounding, as individuals with higher magnesium intake often follow healthier dietary patterns and lifestyles. Although most cohort studies adjust for major confounders, complete elimination of these influences is unlikely, requiring cautious interpretation of causal inference.

Clinical trials assessing supplementation have yielded mixed results. In a 16-week RCT of 63 type 2 diabetes patients with hypomagnesemia, daily magnesium chloride supplementation significantly reduced fasting plasma glucose (8.0 vs. 10.3 mmol/L), hemoglobin A1c (HbA1c) (8.0% vs. 10.1%), and homeostasis model assessment of insulin resistance (HOMA-IR) (3.8 vs. 5.0) compared with placebo [48]. A meta-analysis of 18 RCTs reported improvements in fasting glucose among individuals with type 2 diabetes and in 2 h oral glucose tolerance test (OGTT) glucose among high-risk groups, with a trend toward lower HOMA-IR [49]. Small sample sizes, heterogeneity in magnesium dosage and formulation, and the focus on mainly type 2 diabetes populations collectively limit the precision and generalizability of the pooled findings [49]. By contrast, a double-blind crossover trial in 14 insulin-treated patients with type 2 diabetes and low serum magnesium found that 6 weeks of oral magnesium supplementation (15 mmol/day) increased serum magnesium but did not improve insulin sensitivity, HbA1c, or insulin requirements, apart from a small reduction in high-density lipoprotein (HDL) cholesterol [50]. The limited sample size and short duration of this study make it difficult to draw firm conclusions.

The absence of glycaemic improvement in insulin-treated individuals may reflect the metabolic effects of concurrent insulin therapy, which can alter insulin sensitivity and β-cell responsiveness, potentially masking the benefits of magnesium supplementation [51,52,53]. Evidence suggests that magnesium’s effects are more pronounced in non-insulin-treated individuals, underscoring the importance of treatment context [54]. Moreover, insulin therapy itself may contribute to progressive β-cell dysfunction over time, complicating the interpretation of adjunctive interventions such as magnesium supplementation [52].

Taken together, current trial evidence supports a context-dependent effect, with improvements most evident among individuals with low baseline magnesium or poor glycaemic control, whereas benefits are less consistent in magnesium-replete or intensively insulin-treated populations. This context-specificity highlights the need to identify subgroups most likely to benefit from supplementation rather than assuming uniform metabolic effects. Given the high global burden of type 2 diabetes, monitoring magnesium levels in at-risk groups and ensuring adequate intake through diet or supplementation represent important strategies for prevention and management. Nonetheless, further large-scale, high-quality RCTs are needed to optimize intake recommendations, clarify dose–response relationships, and fully establish the therapeutic potential of magnesium in diabetes care.

### 3.3. Bone and Dental Health

Magnesium is a critical mineral for both skeletal and dental health, acting through multiple mechanisms that support bone mineralization, calcium homeostasis, and periodontal integrity [4,42,44]. In bone tissue, magnesium regulates parathyroid hormone secretion and vitamin D activation, thereby modulating calcium balance, while also directly promoting osteoblast proliferation, differentiation, and function [55]. At the cellular level, magnesium influences the RANK/RANKL/OPG signaling pathway, a central regulator of bone remodeling, thereby maintaining the balance between bone formation and resorption [56].

Observational studies support a protective role of magnesium in skeletal integrity. In a prospective cohort of 73,684 postmenopausal women, higher magnesium intake was associated with greater hip and whole-body bone mineral density [57]. Similarly, in the Osteoarthritis Initiative (*n* = 3765), participants in the highest intake quintile had a significantly lower risk of incident fractures over 8 years [58]. Consistent with these findings, a Finnish long-term prospective cohort of 2245 middle-aged men reported that low serum magnesium concentrations were independently associated with almost a twofold higher risk of total and femoral fractures over 25 years of follow-up [59]. Differences in age, baseline status, co-nutrient exposure (calcium and vitamin D), and mechanical loading likely contribute to variability across observational findings. Collectively, these findings suggest that magnesium status is associated with skeletal integrity across diverse populations, although effect magnitudes vary and do not necessarily imply causality.

Clinical trial evidence is more heterogeneous. In a 12-month RCT of healthy periadolescent girls, 300 mg/day magnesium oxide significantly increased integrated hip bone mineral content, whereas effects on site-specific bone mineral density and biochemical markers were limited [60]. Short-term trials in adults have shown biochemical changes: magnesium citrate (1830 mg/day for 30 days) increased osteocalcin and reduced urinary deoxypyridinoline and intact parathyroid hormone in postmenopausal women [61], and oral magnesium (~365 mg/day for 30 days) suppressed both formation and resorption markers in young men [62]. In contrast, a 12-week RCT in adults with overweight and obesity found no significant effects of combined vitamin D and magnesium supplementation on bone-turnover markers [63]. Trial heterogeneity, particularly choice of magnesium form and dose, intervention duration relative to skeletal turnover, and reliance on biochemical rather than densitometric endpoints, may explain inconsistent effects and suggests magnesium’s role is most plausibly adjunctive to calcium/vitamin D and lifestyle measures.

With regard to dental health, magnesium similarly contributes to the maintenance of alveolar bone and periodontal tissue structure through bone-related pathways and modulation of inflammatory processes [44]. Cross-sectional epidemiological data supports this role. In a population sample of 4290 individuals, higher serum magnesium/calcium ratio among subjects aged 40 years and older was significantly associated with reduced probing depth, less attachment loss, and a greater number of remaining teeth [64]. In matched-pair analyses, magnesium users had less attachment loss and more teeth than non-users [64]. Another study demonstrated that individuals in the highest quartile of magnesium/calcium ratio experienced less progression of periodontal attachment loss over five years [65]. Moreover, in states of systemic inflammation (C-reactive protein > 3 mg/L), high magnesium/calcium ratio was associated with a 40% reduction in tooth loss [65].

Additional evidence from population-based datasets further supports these observations. In NHANES 2013–2014, participants in the highest quintile of dietary magnesium intake had 31% lower odds of periodontitis relative to the lowest quintile [66], and magnesium depletion scores in 8628 United States adults were positively associated with moderate/severe periodontitis [67]. To date, no robust RCTs of magnesium supplementation in otherwise healthy periodontal populations have been documented. The closest evidence comes from a quasi-experimental trial combining magnesium oxide (500 mg/day) and zinc gluconate (50 mg/day) with non-surgical periodontal therapy in patients with type 2 diabetes, which observed improvements in periodontal parameters and oxidative stress markers [68]. Given the observational design of most periodontal studies, residual confounding (diet quality, oral hygiene, smoking) remains a concern, underscoring the need for RCTs to establish causality.

Taking all the aforementioned into account, future well-designed, long-term clinical studies are needed to better define dose–response relationships, to optimize supplementation strategies, and to determine the extent to which magnesium should be incorporated into standard prevention and treatment approaches for both osteoporosis and periodontal diseases. Importantly, it is essential to consider potential risks of excessive intake, as maintaining physiological magnesium balance is fundamental for favorable skeletal and oral outcomes [69].

### 3.4. Neurological and Mental Health

Low magnesium status has been associated with neurological dysfunction, particularly migraines. Magnesium modulates neuronal excitability through voltage-gated calcium antagonism and N-methyl-D-aspartate receptor blockade, thereby attenuating cortical spreading depression and trigeminovascular activation, both core migraine mechanisms [70]. Observational studies have reported lower circulating magnesium concentrations in migraine patients. In a prospective study of adult men and women, mean serum magnesium levels were reduced interictally and further decreased during attacks in patients with migraines, when compared to healthy controls [71]. In an analysis of 10,798 United States adults from NHANES (1999–2004), magnesium intake showed an inverse association with migraine among women, though not in men [72]. Such observational results, however, may be influenced by dietary patterns, sex differences, and unmeasured confounding, which limit causal interpretation.

Clinical trial evidence remains mixed. The MAGraine RCT reported comparable pain reduction with intravenous magnesium sulfate versus prochlorperazine or metoclopramide in emergency settings, suggesting intravenous magnesium as an alternative treatment option for migraines [73]. Another RCT reported that adding 2 g intravenous magnesium sulfate to metoclopramide accelerated early pain relief compared with metoclopramide alone, though the independent effect of magnesium sulfate could not be determined [74]. Variability in study design and co-administered treatments may help explain these inconsistencies. Collectively, acute intravenous studies target nociceptive pathways during attacks, whereas oral prevention trials evaluate longer-term excitability and vascular mechanisms. This difference in therapeutic context likely contributes to mixed findings. For prevention, professional guidelines suggest oral magnesium (commonly 400–500 mg/day magnesium oxide) may be offered, with gastrointestinal intolerance being the primary dose-limiting factor [75,76].

Beyond migraine, higher habitual magnesium intake has been associated with larger brain volumes and fewer white-matter lesions in UK Biobank participants, particularly among women [77]. Follow-up analyses indicated that, above 350 mg/day, each additional 1 mg/day of magnesium intake was related to 0.0105% larger gray matter, 0.0122% larger white matter, and 0.002% larger right hippocampal volume [78]. These findings indicate that habitual magnesium intake may contribute to structural brain integrity, particularly in aging populations, although the extent to which this reflects a direct neuroprotective effect versus broader dietary and lifestyle patterns requires further clarification. Given the observational design, residual confounding and reverse causation cannot be excluded.

Lower serum magnesium concentrations have been reported in Alzheimer’s disease and mild cognitive impairment, and experimental studies suggest effects on synaptic plasticity and amyloid-β deposition [79,80]. An 8-week open-label trial of L-threonic acid magnesium salt in patients with mild to moderate Alzheimer’s disease showed modest cognitive improvement, although interpretation is limited by lack of placebo control and small sample size [81]. Disturbances in magnesium transport and homeostasis have also been described in Parkinson’s disease [82,83], but robust RCTs are lacking. Overall, the evidence base for magnesium in neurodegenerative disorders remains preliminary and is mainly limited by study design constraints.

Magnesium status also appears relevant to mental health. Low dietary or serum magnesium has been associated with higher prevalence and severity of depression and anxiety symptoms, findings supported by recent reviews [84,85]. Mechanistic studies support this association, as magnesium appears to modulate stress-response pathways and neurotransmitter balance, including promoting GABAergic inhibition while tempering excitatory glutamatergic signaling [8].

Across cognitive and mood outcomes, effect estimates appear modest and variable, with signals strongest in subgroups with low intake/status or higher symptom burden and less consistent in replete, low-symptom cohorts. Interventional studies have tested whether magnesium supplementation can alleviate mood symptoms. A recent meta-analysis of RCTs including 325 participants found that magnesium supplementation reduced depressive symptoms compared with control, although the evidence base remains small and heterogeneous [86]. Individual trials have reported improvements in depressive symptoms with magnesium supplementation [87,88,89], though some studies did not confirm these effects [90]. Differences in baseline magnesium status, symptom severity, dosing, and study duration likely contribute to inconsistent results.

Magnesium may also influence sleep quality. Observational studies consistently associate higher magnesium intake with longer sleep duration and fewer awakenings; however, RCTs have shown inconsistent effects of supplementation on sleep disorders [91]. More recent small-scale studies reported improvements in sleep quality following magnesium supplementation, but formulations and outcome measures varied [92,93]. The limited sample sizes and variability in study approaches complicate firm conclusions.

Taken together, mechanistic, observational, and preliminary clinical evidence suggests that magnesium may exert beneficial effects across domains of neurological, cognitive, mood, and sleep health. However, current clinical trial evidence remains limited in size, duration, and methodological rigor. At present, maintaining adequate magnesium status through diet or supplementation appears reasonable, but therapeutic claims for neurological and mental disorders should remain cautious until larger, well-controlled, long-term randomized trials clarify optimal dosing, formulations, target populations, and magnitude of benefit.

### 3.5. Chronic Inflammation

Magnesium deficiency is increasingly recognized as a contributor to chronic low-grade inflammation, a key driver of many noncommunicable diseases. Low magnesium status promotes mitochondrial dysfunction, oxidative stress, and nuclear factor kappa B (NF-κB) activation, increasing the production of pro-inflammatory cytokines such as interleukin (IL)-1β, IL-6, and tumor necrosis factor alpha (TNF-α) [94,95,96]. It also impairs endothelial function and immune regulation, thereby fostering vascular and metabolic inflammatory processes [96,97].

Epidemiological evidence supports these associations. In the UK Biobank, higher dietary magnesium intake was inversely associated with inflammatory markers, including C-reactive protein, leukocyte count, and glycoprotein acetylation [78]. Recent review further emphasizes that magnesium deficiency affects both innate and adaptive immune responses, predisposing individuals to persistent inflammation [98].

The relationship between chronic inflammation and magnesium status has direct implications for cardiovascular disease development. Low magnesium status not only promotes inflammatory signaling, but this sustained inflammatory environment accelerates vascular injury and cardiometabolic deterioration over time, thereby linking inadequate intake with increased cardiovascular risk [37]. This vulnerability is particularly evident in populations with chronic kidney disease, who demonstrate lower magnesium status, higher inflammatory biomarker levels, and greater cardiovascular complications [99]. This suggests a bidirectional interaction in which chronic inflammatory states may further deplete magnesium stores, thereby amplifying disease progression. Clinical trials provide preliminary support for anti-inflammatory effects of magnesium supplementation. A meta-analysis of nine RCTs in overweight and obese adults reported significant reductions in high-sensitivity C-reactive protein (–1.19 mg/L) and TNF-α (–0.87 pg/mL) with magnesium and vitamin D supplementation, while results for IL-6 were less consistent [100]. Observed reductions in C-reactive protein levels are modest and often occur alongside co-supplementation (e.g., vitamin D), which complicates attribution and highlights the need for factorial or monotherapy designs. Experimental work also shows that magnesium enhances regulatory T cell activity and reduces arthritis severity in animal models, pointing to potential immunomodulatory benefits [101].

Current evidence suggests that insufficient magnesium intake contributes to systemic inflammation and may influence the progression of chronic disease. Improving magnesium status through dietary change or supplementation may help reduce inflammatory burden in at-risk populations, but further large-scale RCTs are required to clarify dose–response relationships and determine long-term clinical impact.

### 3.6. Cancer

Magnesium status has been linked to cancer risk, progression, and mortality, with the strongest and most consistent evidence currently available for colorectal cancer (CRC). As a mineral essential for DNA repair, oxidative stress regulation, and genomic stability, magnesium may influence colorectal carcinogenesis. Epidemiological, cohort, and case–control studies consistently report a modest but significant inverse association between dietary magnesium intake and CRC incidence, although effect sizes vary by population, sex, and genetic background. A meta-analysis of eight prospective studies found that individuals in the highest intake category had a relative risk of 0.89 for CRC compared with the lowest intake [102]. Similarly, a large Japanese cohort reported a hazard ratio of 0.65 among men in the highest quintile of intake [103], while a Korean case–control study reported an odds ratio of 0.65, with the effect modulated by the *INSR* rs1799817 genetic variant, suggesting gene-nutrient interactions [104]. Furthermore, in a Swedish cohort of women, those with the highest magnesium intake demonstrated a relative risk of 0.59 for CRC compared to the lowest intake group, indicating a significant reduction in risk [105]. CRC likely shows the strongest and most consistent signal because the colon is directly exposed to luminal magnesium, and magnesium–microbiota–epithelial interactions may amplify biological relevance compared with more distal cancer sites.

More broadly, adequate magnesium contributes to genomic stability by supporting DNA repair and reducing oxidative damage. Experimental studies show that optimal magnesium levels protect cells against free radical–induced DNA damage and mutation, while deficiency impairs DNA repair pathways and increases oxidative stress, creating a permissive environment for malignant transformation [98,106,107]. However, much of the epidemiological evidence is observational and residual confounding by diet quality, lifestyle, and screening practices cannot be excluded.

Beyond CRC, epidemiological data also suggests broader associations with cancer outcomes. A dose–response meta-analysis of prospective studies found that each additional 100 mg/day increase in dietary magnesium intake was associated with a 5% reduction in cancer mortality [108]. In a prospective United States study, individuals in the highest tertile of total magnesium intake had a 56% lower incidence of primary liver cancer (hazard ratio: 0.44; 95% confidence interval: 0.24–0.80) and lower mortality risk (subdistribution hazard ratio: 0.37; 95% confidence interval: 0.19–0.71), when compared with those in the lowest tertile [109]. Low dietary magnesium has further been associated with impaired DNA repair capacity and increased lung cancer risk [110].

Interventional evidence remains preliminary. In a precision-based RCT, magnesium supplementation modified gut microbiota in a genotype-dependent manner, with exploratory links to colorectal polyp recurrence, though findings were inconsistent and not uniformly significant [111]. In oncology treatment settings, a recent multi-center retrospective study reported that higher serum magnesium concentrations were associated with longer progression-free and overall survival in patients treated with immune checkpoint inhibitors [112]. At present, causal inference remains limited by the scarcity of randomized prevention trials and by heterogeneity in exposure assessment, reinforcing the need for prospective studies with standardized magnesium metrics and adjudicated cancer endpoints.

Overall, magnesium appears to have a protective role in carcinogenesis, with the clearest and most consistent evidence available for CRC, and additional emerging data in other cancer types. Large-scale, well-designed studies are needed to clarify causality, define dose–response relationships, and determine whether magnesium has an actionable therapeutic role in oncology.

### 3.7. Pregnancy and Reproductive Health

A recent review noted that magnesium deficiency in pregnancy may be linked to adverse maternal and fetal outcomes, though evidence remains heterogeneous [113].

Magnesium supplementation has shown mixed results in obstetric practice. Large RCTs and meta-analyses confirmed that intravenous magnesium sulfate prevents seizures in women with severe preeclampsia and reduces the risk of cerebral palsy in very preterm infants, without increasing pediatric mortality [114,115,116]. Regarding the prevention of preeclampsia, a meta-analysis of seven RCTs including 2653 pregnant women concluded that oral magnesium supplementation may reduce the risk of preeclampsia, particularly in those at high risk [117]. In women with gestational diabetes, a meta-analysis of four small RCTs (*n* = 198) reported improvements in fasting plasma glucose and fasting insulin with magnesium supplementation [118]. On the other hand, a meta-analysis of oral magnesium for pregnancy-related leg cramps concluded that supplementation did not significantly reduce symptoms [119]. Overall, magnesium sulfate is established for seizure prophylaxis in severe preeclampsia, whereas preventive oral supplementation shows promise primarily in higher-risk or deficient populations, with limited evidence for routine use in uncomplicated pregnancies. Heterogeneity across trials likely reflects differences in baseline magnesium status, underlying obstetric phenotype, and supplementation timing (preconception vs. late gestation), suggesting that intervention effectiveness is context-dependent rather than uniform.

Beyond pregnancy, magnesium has been studied in reproductive disorders such as polycystic ovary syndrome (PCOS). In PCOS, magnesium alone showed no significant benefits, while co-supplementation with other nutrients has produced variable metabolic improvements [120,121].

Future trials should clarify dose–response relationships, long-term safety, and whether supplementation offers preventive or adjunctive value in high-risk pregnancy and reproductive disorders.

### 3.8. Sarcopenia and Muscle Function

Adequate magnesium is important for normal muscle function, whereas deficiency contributes to muscle weakness, cramps, and impaired physical performance [122].

Observational evidence consistently links magnesium status to muscle strength and sarcopenia risk in older adults. In the prospective epidemiological study (*n* = 1138; mean age 66.7 years), higher serum magnesium was independently associated with greater grip strength, leg muscle power, knee extension torque, and ankle extension strength [123]. Cross-sectional analyses from population-based cohorts have reported that higher dietary magnesium intake is associated with greater muscle mass and strength and with lower prevalence of sarcopenia [124,125].

RCT data on magnesium supplementation and muscle outcomes are limited. In a RCT of 139 healthy older women, 300 mg/day magnesium for 12 weeks improved Short Physical Performance Battery scores, chair-stand time, and 4 min walking speed versus control, with no serious adverse events [126]. Beyond this study, evidence in older adults is sparse, and findings from small trials have not consistently shown gains in muscle mass or strength [127]. Differences in baseline status, concomitant protein intake, and the presence or absence of structured resistance exercise likely influence outcomes, with functional performance measures appearing more responsive than lean-mass endpoints over short interventions.

Taken together, current evidence suggests that magnesium status is an important determinant of muscle health in aging populations. Supplementation may enhance physical performance, particularly in those with low baseline intake, but results remain inconsistent. Future large-scale, long-term RCTs are needed to clarify dose–response effects and to determine whether magnesium supplementation should be integrated into sarcopenia prevention and treatment strategies.

Magnesium thus affects multiple physiological systems, with clinical implications extending across cardiometabolic, musculoskeletal, neurological, inflammatory, reproductive, and oncologic health domains. Whether these effects translate into meaningful benefit at the population level depends not only on biological plausibility and intervention evidence, but also on accurate characterization of magnesium status. Consequently, understanding how deficiency is defined, how common it is, and what factors determine risk becomes essential for contextualizing the clinical relevance of the associations described above.

## 4. Magnesium Status Worldwide: Biomarkers, Prevalence, and Population Determinants

### 4.1. Biomarkers of Magnesium Status

The assessment of magnesium status remains challenging due to the lack of a single, universally reliable biomarkers. Approximately 1% of total body magnesium is located in blood compartments, while the majority is stored in bone and soft tissues, which substantially limits the diagnostic accuracy of serum-based indicators [128]. This uneven distribution contributes to inconsistent classification of deficiency in both clinical and epidemiological research.

Serum total magnesium (tMg) is the most frequently used biomarker due to its accessibility and established reference ranges. However, tMg reflects only a very small fraction of total body magnesium (less than 1%) and is strongly influenced by albumin binding, which can lead to misclassification in conditions such as hypoalbuminemia [122,129,130,131]. Increasing evidence suggests that values in the low-normal range (<0.85 mmol/L) may be associated with greater cardiometabolic risk, prompting proposals for a higher lower-limit threshold, and indicating that current hypomagnesemia cut-offs may underestimate true deficiency [130,132]. Serum tMg is further affected by renal function and does not accurately reflect intracellular stores [133,134].

Ionized magnesium (iMg), which represents the physiologically active fraction, is increasingly recognized as a more informative indicator, particularly in acute and critical care settings. Its clinical utility, however, is limited by availability and a lack of standardization [135,136]. Red blood cell magnesium may better reflect intracellular status, but considerable inter-individual variability restricts its diagnostic utility [137].

Twenty-four-hour urinary magnesium excretion provides insight into recent dietary intake and renal handling. Metabolic balance studies have shown that when daily magnesium intake is below 250 mg, urinary excretion generally falls within 40–80 mg/day. At higher intakes, above 250 mg/day, urinary losses typically rise to 80–160 mg/day [138]. Comparable excretion values have also been observed in healthy adults consuming their usual diets [1]. Despite these insights, urinary measures alone are insufficient for definitive assessment.

Historically, the magnesium tolerance test was considered a functional gold standard, capable of detecting intracellular deficiency, but its invasiveness, cost, and logistical complexity limit routine use [139]. These limitations have stimulated interest in functional biomarkers that capture downstream physiological effects. Enzymatic and inflammatory markers such as Na/K ATPase, thromboxane B_2_, C-reactive protein, and endothelin-1 have shown potential to detect early alterations in magnesium-dependent pathways, but require further validation [6,139].

Given these limitations across individual indicators, no single measure is sufficient to characterize magnesium status in all contexts. A practical approach combines dietary intake assessment with serum tMg and urinary magnesium excretion [129], while emerging functional markers offer a promising direction for future refinement. However, standardized methodologies and consensus reference values are needed before these measures can be applied reliably in population surveillance or clinical decision-making. In practice, most large surveys continue to rely on dietary assessment and serum tMg, with selective use of iMg and urinary indices restricted to research or specialized clinical settings. A detailed comparative summary of the main biomarkers, their reference ranges, and practical considerations is provided in Table 1.

### 4.2. Dietary Requirements

Dietary reference values for magnesium differ across regions. In the United States and Canada, the Recommended Dietary Allowance (RDA) is 400–420 mg/day for adult men and 310–320 mg/day for adult women, with modest increases during pregnancy (350–360 mg/day) and lactation (310–320 mg/day) [129]. On the other hand, the European Food Safety Authority (EFSA) sets Adequate Intakes (AI) at 350 mg/day for men and 300 mg/day for women [143]. However, metabolic balance data from controlled feeding studies indicate that neutral magnesium balance in healthy adults was achieved at considerably lower intakes (165 mg/day) [144]. Because such studies capture true physiological retention and excretion dynamics under standardized dietary conditions, they may better reflect minimal physiological requirements than population reference values derived from epidemiological intake distributions. This suggests that current population-based reference values may not fully align with physiologically determined needs and reinforces the importance of contextual interpretation when determining optimal intake.

The elderly population represents a physiologically vulnerable group with increased magnesium needs. Age-related declines in intestinal absorption and increases in renal magnesium loss contribute to lower systemic magnesium levels [145,146]. Observational studies show that inadequate intake in this population is linked to chronic inflammation and frailty [147,148]. Higher dietary intake has been associated with reduced frailty risk, particularly in men, highlighting the need for targeted strategies in aging populations [148].

Pregnancy and early neonatal life also represent periods of increased magnesium demand. Enhanced renal excretion and fetal growth increase maternal requirements, yet magnesium deficiency remains highly prevalent worldwide [149,150]. Insufficient maternal intake is linked to complications such as preterm birth, gestational diabetes, and placental dysfunction, while low neonatal magnesium is associated with impaired postnatal adaptation and poorer Apgar scores [151,152]. Ensuring adequate intake during pregnancy is therefore critical to support both maternal and neonatal outcomes.

Gender-specific differences are increasingly recognized. In women, higher magnesium intake is associated with improved bone density and reduced risk of cognitive decline in later life [153,154]. By contrast, men may require higher intakes to reduce frailty risk, although bone health benefits appear to plateau at higher intakes [148,154]. During perimenopause, magnesium has been implicated in the alleviation of vasomotor, mood, and sleep symptoms, while also contributing to cardiovascular and bone health [155,156,157].

Finally, athletes and individuals with high physical activity levels often have increased magnesium requirements due to losses through sweat and urine. Estimates suggest that needs may be 10–20% higher compared with sedentary individuals [158]. Since magnesium plays a pivotal role in energy metabolism and muscular function, supplementation has been linked to improvements in performance parameters, though benefits appear most consistent when correcting deficiency rather than providing excess [158,159,160].

The above-mentioned findings indicate that while RDAs and AIs provide population-level guidance, optimal magnesium intake should be tailored to age, sex, reproductive status, training load, and clinical context. Such individualized approaches may help prevent deficiency and optimize health outcomes in vulnerable subgroups. Additionally, data from controlled metabolic balance studies indicate that body size may modestly influence magnesium requirements, and should therefore be considered when evaluating individual intake adequacy [144].

### 4.3. Determinants and Global Patterns of Magnesium Deficiency

Suboptimal magnesium intake is widespread across all world regions, though the magnitude varies depending on dietary patterns, socioeconomic status, and the quality of national surveillance systems [143,161,162,163,164].

Large population surveys such as the United States NHANES and the Chinese National Nutrition Survey consistently demonstrate that a substantial proportion of adults fail to meet estimated average requirements for magnesium [161,162]. In Europe, EFSA and United Kingdom National Diet and Nutrition Survey reports similarly indicate that measurable fractions of adults, particularly women and older individuals, consume less than national reference values [143,165].

Biochemical hypomagnesemia appears relatively infrequent in community settings, but prevalence increases substantially in hospitalized and critically ill patients due to altered renal handling, inflammation, medication exposure, and acute metabolic stress. However, these prevalence estimates are based on traditional cut-offs (<0.75 mmol/L) and do not account for individuals with serum magnesium values within the low-normal range (e.g., <0.85 mmol/L), who may still have inadequate magnesium status [166,167]. Data from the Canadian Health Measures Survey indicate that 9.5–16.6% of adults have serum magnesium concentrations below 0.75 mmol/L [168]. Similar findings were observed in a large unselected German population, where 14.5% of participants had hypomagnesemia [169]. In contrast, prevalence in hospitalized and intensive care unit populations is markedly higher, with studies reporting 20–60% of patients affected, depending on clinical setting and diagnostic thresholds [140,170,171].

Despite regional variation in intake and biochemical deficiency rates, multiple biological, dietary, and environmental factors collectively determine magnesium status. Understanding these determinants is essential for identifying at-risk populations and developing targeted interventions. Table 2 summarizes the principal contributors to magnesium deficiency in the general population.

These global patterns indicate that insufficient intake, rather than isolated disorders of absorption or renal handling, is the primary driver of magnesium deficiency worldwide. Consequently, understanding which foods and waters contribute meaningfully to intake, and which dietary and processing factors shape bioaccessibility, becomes central for developing population strategies capable of sustainably reducing deficiency burden.

## 5. Dietary Sources of Magnesium

Magnesium is widely distributed across diverse food groups, with plant-based foods contributing the majority of dietary intake in most populations. The richest natural sources include nuts, seeds, legumes, whole grains, and green leafy vegetables, all of which provide additional nutrients such as fiber, potassium, and polyphenols that collectively support cardiometabolic health [184,185]. Animal-derived foods, including dairy products and fish, as well as fortified products such as breakfast cereals and plant-based milk alternatives, make meaningful contributions to total intake [4].

Drinking water represents an additional, though often overlooked, source of magnesium. The concentration of magnesium in water varies substantially depending on geological conditions and water treatment processes [186]. Consumption of magnesium-rich mineral water has been associated with improved magnesium status and favorable vascular outcomes, likely through enhanced bioavailability and the mineral’s vasodilatory properties [187,188]. Such waters may therefore serve as a useful adjunct in populations with suboptimal intake.

Bioavailability of dietary magnesium is influenced by several factors. Compounds such as phytates and oxalates can chelate magnesium and reduce intestinal absorption [182,183], whereas fermentation, soaking, or sprouting of plant foods may improve its bioaccessibility [189,190].

Variability in both food composition and matrix effects means that magnesium intake is not solely a function of total quantity consumed, but also of food choice and preparation practices. Therefore, strategies aiming to improve magnesium status should emphasize dietary pattern quality rather than isolated single-food additions, with particular attention to plant-rich diets, minimally processed grains, and traditional food preparation techniques that enhance mineral bioaccessibility.

Although dietary intake remains the primary determinant of magnesium status, real-world intake data indicate that a substantial proportion of adults do not achieve recommended targets through food alone. In these situations, supplemental magnesium becomes an important complementary strategy, particularly for individuals with elevated physiological needs or limited dietary access.

## 6. Magnesium Supplementation: Bioavailability and Safety

Magnesium supplementation is an established strategy for correcting deficiency and supporting physiological functions when dietary intake alone is insufficient. The choice of preparation and dose depends on the severity of depletion and the individual’s clinical condition. Mild magnesium deficiency is generally managed with oral supplementation providing 300–600 mg elemental magnesium per day, whereas acute or severe hypomagnesemia requires parenteral administration under medical supervision [191,192]. A comprehensive approach is recommended, prioritizing dietary improvement and lifestyle modification alongside pharmacological correction.

### 6.1. Bioavailability of Magnesium Compounds

The efficacy of supplementation depends largely on the chemical form of magnesium and its bioavailability. Magnesium is absorbed in the intestine via both passive paracellular diffusion and active transcellular transport. Organic salts, such as magnesium citrate, glycinate, and malate, show higher solubility and absorption efficiency compared with inorganic forms like magnesium oxide or sulfate [193,194].

Human trials have consistently demonstrated that magnesium citrate produces higher serum and urinary magnesium levels than magnesium oxide in both acute and chronic supplementation [194,195]. Magnesium glycinate exhibits particularly high absorption and gastrointestinal tolerance, attributed to its chelated structure that facilitates transport and minimizes interaction with other minerals [92,196]. Magnesium malate may additionally support energy metabolism and muscle performance, while magnesium taurate and orotate have been investigated for cardiovascular support, and magnesium threonate for neurocognitive function [197,198,199].

Inorganic salts such as magnesium oxide, hydroxide, and sulfate are less bioavailable than most organic forms, but adverse gastrointestinal effects are mainly seen when these forms are used at pharmacological doses for the purpose of alleviating constipation. At reasonable doses, they can be absorbed sufficiently to contribute to magnesium intake without necessarily producing laxative effects [193,200,201,202]. These forms may therefore be suitable for gastrointestinal use but may be less efficient for systemic repletion. Overall, differences in bioavailability across formulations indicate that supplement selection should be guided by clinical purpose, baseline magnesium status, and tolerance rather than by chemical classification alone. This underscores the need for formulation-specific evaluation within defined therapeutic contexts rather than assuming uniform interchangeability of supplemental magnesium forms. Table 3 summarizes the most commonly used magnesium formulations, highlighting their chemical type, relative bioavailability, principal clinical applications, and potential adverse effects.

### 6.2. Safety and Tolerability

The Tolerable Upper Intake Level (UL) for supplemental magnesium is 350 mg/day (excluding food sources) in adults [129]. Intakes above this threshold may cause diarrhea, nausea, or hypotension [195]. However, magnesium from food has no known adverse effects in healthy individuals. Supplement selection should therefore balance bioavailability, safety, and affordability—with citrate and glycinate preferred for efficiency and tolerance, and oxide serving as an accessible, low-cost option particularly when a laxative effect is desired.

Excessive magnesium intake can lead to hypermagnesemia, especially in individuals with impaired renal function due to reduced excretory capacity [210]. Recent evidence suggests that those with an estimated glomerular filtration rate ≤43.1 mL/min are particularly vulnerable [211]. Nevertheless, the evidence base regarding magnesium safety in renal impairment remains limited and heterogeneous. Many available studies involve small or clinically diverse populations and often lack standardized assessment of confounders such as dietary magnesium intake, concomitant medications, and dialysis parameters [212]. Meta-analyses in this area may also be affected by methodological variability and publication bias, potentially obscuring dose–response relationships and underestimating adverse event rates. While adherence to established dietary reference intakes minimizes toxicity risk in the general population, cautious interpretation of aggregated evidence and individualized monitoring are warranted in patients with compromised renal function.

Although the choice of supplemental preparation and dose influences the extent to which magnesium status can be corrected, the biological effects of magnesium ultimately depend on how it integrates within the wider nutrient network. Magnesium does not act in isolation; it modifies, and is modified by, calcium, potassium, trace elements, and key vitamin-dependent pathways that govern neuromuscular excitability, energy metabolism, bone mineralization, and systemic homeostasis. Understanding these nutrient–nutrient interactions is therefore essential to interpret both the clinical effects of supplementation and the variability in response across individuals.

## 7. The Central Role of Magnesium in Nutrient Metabolism

Magnesium’s influence extends across calcium, potassium, zinc, copper, and vitamin D metabolism [122,213].

### 7.1. Magnesium, Calcium, and Vitamin D Synergy

One of magnesium’s key physiological roles is the modulation of calcium homeostasis. Acting as a natural calcium antagonist, magnesium prevents intracellular calcium overload, thereby preserving neuromuscular coordination and cardiac rhythm stability [42,214]. Optimal bone strength depends on the coordinated action of magnesium, calcium, and vitamin D, which together regulate bone mineralization and remodeling [215,216].

Magnesium is essential for the enzymatic activation of vitamin D [217]. It serves as a cofactor in both hepatic 25-hydroxylation and renal 1α-hydroxylation, the sequential steps converting vitamin D into its biologically active form, 1,25-dihydroxyvitamin D [213,218]. Deficiency of magnesium can impair these conversions, limiting calcium absorption and contributing to skeletal and cardiovascular complications. Conversely, vitamin D enhances intestinal calcium absorption—primarily in the duodenum and jejunum—and may modestly facilitate magnesium uptake [218,219]. This bidirectional relationship highlights magnesium’s essential role in maintaining the vitamin D–calcium axis.

### 7.2. Interactions with Potassium and Trace Elements

Magnesium also plays an important role in potassium regulation by maintaining the activity of the Na^+^/K^+^-ATPase pump and preventing intracellular potassium loss [42,214]. Deficiency of magnesium leads to secondary hypokalemia and neuromuscular irritability, manifesting as tremors, cramps, or muscle weakness [220,221]. Through these mechanisms, magnesium stabilizes neuronal membranes and supports efficient nerve transmission [79].

In addition, magnesium interacts closely with zinc and copper, elements that together participate in hundreds of enzymatic reactions. Both magnesium and zinc serve as cofactors for enzymes involved in ATP synthesis and nucleic-acid metabolism [222], while copper supports antioxidant defense through the activity of superoxide dismutase [223]. Imbalances in one mineral can influence the absorption or function of others, emphasizing the need for balanced intake and careful consideration of supplement combinations [224,225].

### 7.3. Magnesium and B Vitamins: Neuro-Metabolic Synergy

Beyond its interactions with minerals and vitamin D, magnesium also acts synergistically with vitamin B6 and vitamin B12 in pathways critical for neurological function and energy metabolism. Vitamin B6 enhances the cellular uptake and intracellular retention of magnesium, thereby potentiating its metabolic and neuromuscular effects [226]. Clinical studies indicate that combined supplementation of magnesium and vitamin B6 can reduce premenstrual symptoms such as nervous tension, irritability, anxiety, and mood swings [227]. Furthermore, oral magnesium supplementation has been shown to alleviate stress in adults with low serum magnesium, with the combination of magnesium and vitamin B6 offering additional benefit among individuals experiencing severe or extreme stress [228].

Vitamin B12 complements this interaction by participating in methylation reactions, myelin synthesis, and red blood cell formation, processes that depend indirectly on adequate magnesium for ATP generation and enzymatic activation [229]. Collectively, these nutrients support nervous system stability, mood regulation, and metabolic homeostasis, underscoring the importance of combined micronutrient sufficiency in maintaining optimal neuro-metabolic health.

The wide range of nutrient interactions demonstrates that magnesium status influences far more than individual metabolic pathways. Because magnesium contributes simultaneously to cardiovascular regulation, glucose control, bone remodeling, neuromuscular function, immune activity, and genomic stability, insufficient intake creates systemic vulnerability across multiple organ systems. This positions magnesium deficiency not as a narrow micronutrient problem, but as a broader public health issue with implications for chronic disease prevention and population health management.

## 8. Magnesium Deficiency: A Public Health Challenge

Despite extensive evidence of magnesium’s physiological importance, population-level deficiency remains a persistent and underrecognized public health challenge. Recent global analyses indicate that up to one-third of adults fail to meet recommended magnesium intakes, contributing to increased vulnerability to cardiometabolic and inflammatory disorders [4,44,95]. During the COVID-19 pandemic, hypomagnesemia was further associated with impaired immune response and adverse clinical outcomes, underscoring its relevance beyond chronic disease [230,231].

While dietary modification remains the foundation of magnesium sufficiency, complementary population-level interventions such as food and water fortification and targeted supplementation programs have gained attention as practical strategies to close persistent intake gaps [44]. These efforts are most effective when aligned with broader nutrition and health promotion frameworks that address dietary quality, food accessibility, and socioeconomic disparities [232]. Importantly, the persistence of magnesium insufficiency in high-income as well as low-income settings indicates that the problem is not solely dietary, but reflects broader structural determinants of food systems, dietary patterns, and chronic disease trajectories.

Incorporating magnesium monitoring into national nutrition surveillance could help identify at-risk populations and guide evidence-based policy adjustments. Emerging personalized approaches, integrating dietary assessment, comorbidity profiles, and genetic predispositions, may further optimize supplementation strategies [4].

Furthermore, advances in magnesium assessment hold promise for improving public health surveillance and clinical management. While routine serum measurements remain widely used, future progress will require the refinement and standardization of physiologically meaningful indicators beyond tMg, which may enhance diagnostic accuracy in both population monitoring and high-risk clinical settings [233]. Integrating these improved assessment tools into national screening frameworks would enable earlier detection and more targeted prevention of deficiency.

A summary of population-level strategies for addressing magnesium deficiency is presented in Table 4.

## 9. Conclusions

Magnesium deficiency remains a prevalent yet underrecognized contributor to global morbidity. Inadequate magnesium status is linked to a wide range of chronic health conditions, including hypertension, cardiovascular disease, diabetes, osteoporosis, and mood disorders.

Despite this broad physiological relevance, magnesium deficiency remains insufficiently prioritized in clinical and public health practice. Increasing population awareness, integrating magnesium within nutrition policy frameworks, and improving routine monitoring strategies are essential steps toward earlier risk identification and better prevention.

Sustainable prevention requires coordinated strategies that combine dietary education, food fortification, targeted supplementation, and personalized approaches that account for metabolic phenotype, sex, age, body size, comorbidities, and genetic variability. Strengthening these measures could substantially reduce the burden of noncommunicable diseases and improve population health resilience.

Looking forward, research should prioritize defining physiologically relevant intake thresholds across diverse phenotypes, clarifying dose–response relationships, and addressing the heterogeneity observed in clinical trials, especially through stratification based on baseline status. A deeper mechanistic understanding of nutrient–nutrient interactions, particularly within the calcium–vitamin D–magnesium axis, is needed to more precisely guide both clinical recommendations and policy development. Establishing these foundations will allow the field to move beyond descriptive associations toward more predictive, targeted, and translationally actionable strategies for magnesium in cardiometabolic and population health.

## Figures and Tables

**Table 1 nutrients-17-03626-t001:** Overview of biomarkers for assessing magnesium status.

Biomarker	Biological Domain Assessed	Reference Values	Advantages	Limitations	References
tMg	extracellular pool	0.75–0.95 mmol/L (typical); <0.85 mmol/L linked to cardiometabolic risk; ≥0.85 mmol/L proposed cut-off	widely available; inexpensive	<1% body Mg; albumin-dependent; weak storage marker	[122,129,130,131,132]
iMg	bioactive free fraction	0.40–0.60 mmol/L (assay-dependent)	stronger clinical correlations	limited availability; technical and standardization challenges	[135,136]
24 h urinary Mg	intake and renal handling	40–80 mg/day (<250 mg intake); 80–160 mg/day (>250 mg intake); adult reference range: 12–255 mg/day; >24 mg/day in hypomagnesemia indicated renal wasting	differentiates inadequate intake vs. renal loss	time-consuming; intake-dependent	[131,140,141,142]
Dietary assessment	usual intake	compared to dietary reference values	identifies modifiable dietary risk factors; feasible for population studies	recall bias; high day-to-day variability	[129,143]

Abbreviations: tMg, serum total magnesium; iMg, ionized magnesium; Mg, magnesium.

**Table 2 nutrients-17-03626-t002:** Key determinants and risk factors influencing magnesium status in the general population.

Determinant	Pathway	Effect on Magnesium Status	References
Dietary intake	low consumption of magnesium-rich foods, high intake of processed foods	decrease in total body Mg and serum Mg	[2,172]
Age	lower intestinal absorption efficiency, higher renal excretion	older adults at higher risk of deficiency	[145,146]
Sex/Reproductive status	hormonal and metabolic differences; pregnancy increases requirements	women, especially pregnant and postmenopausal, more likely to have low Mg	[143,149,173]
Socioeconomic status	diet quality, food accessibility	lower socioeconomic status associated with lower Mg intake	[174,175]
Chronic diseases	diabetes, hypertension, chronic kidney disease, gastrointestinal disorders increase urinary or intestinal Mg losses	higher Mg loss or redistribution leading to biochemical deficiency	[176,177]
Medications	diuretics, proton pump inhibitors, certain antibiotics	lower renal reabsorption or intestinal absorption	[30,178]
Alcohol consumption	renal Mg wasting, malnutrition	lower serum Mg and intracellular Mg	[179,180]
Physical activity	sweat and urinary Mg loss	mild decrease in Mg status if intake not increased	[181]
Dietary inhibitors	phytates, oxalates	lower intestinal absorption	[182,183]
Comorbid micronutrient deficiencies	interactions with calcium, vitamin D, potassium	deficiency in Ca or vitamin D may exacerbate Mg deficiency	[172]

Abbreviations: Mg, magnesium; Ca, calcium.

**Table 3 nutrients-17-03626-t003:** Overview of magnesium supplement formulations with their bioavailability, therapeutic uses, and safety profile.

Magnesium Form	Type	Bioavailability	Common Applications	Typical Adverse Effects	References
Magnesium citrate	Organic	High	General supplementation, fatigue, constipation, muscle cramps	Loose stools at high doses	[193,194]
Magnesium glycinate (bisglycinate)	Organic	Very high	Stress reduction, sleep, muscle relaxation, sensitive stomach	Generally well tolerated	[92,196]
Magnesium malate	Organic	High	Fatigue, muscle pain, athletic recovery	Generally well tolerated	[203,204]
Magnesium taurate	Organic	High	Cardiovascular and metabolic support	Generally well tolerated	[33]
Magnesium orotate	Organic	High	Heart health, energy metabolism	Expensive, generally well tolerated	[92]
Magnesium threonate	Organic	High	Cognitive and neurological support	Gastrointestinal discomfort, rarely lethargy	[198,199,203,205,206]
Magnesium ascorbate	Organic	High	Combined magnesium and vitamin C supplementation, immune support	Mild gastrointestinal discomfort	[207]
Magnesium lactate	Organic	Moderate-high	General supplementation	Mild gastrointestinal discomfort	[208]
Magnesium aspartate	Organic	High	Metabolic and electrolyte balance	Mild gastrointestinal discomfort	[208]
Magnesium oxide	Inorganic	Low	Pharmacological use for constipation; may contribute nutritionally at reasonable doses	Loose stools, gas mainly at higher pharmacological doses	[194]
Magnesium hydroxide	Inorganic	Low	Pharmacological laxative use	Loose stools mainly at higher pharmacological doses	[200]
Magnesium sulfate	Inorganic	low (oral); high (intravenous)	Intravenous therapy, baths for muscle relaxation, oral use primarily pharmacological laxative	Strong laxative effect orally only at pharmacological doses	[200,201]
Magnesium chloride	Inorganic	Moderate-high	General supplementation, topical preparations	Skin irritation (in topical form), gastrointestinal upset possible at higher doses	[209]

**Table 4 nutrients-17-03626-t004:** Public health strategies for the prevention and management of magnesium deficiency [42,229,234,235].

Strategy	Description	Target Groups	Advantages	Limitations
Nutritional education	Promotion of magnesium-rich diets through public campaigns, dietary guidelines, and school-based programs.	General population, students, caregivers.	Improvement in long-term dietary habits and health literacy.	Requires sustained engagement, effectiveness depends on food access.
Food and water fortification	Addition of magnesium to staple foods (e.g., flour, milk) or drinking water to enhance population intake.	General population, particularly regions with low magnesium intake.	Cost-effective, equitable, wide population reach.	Needs regulatory oversight, monitoring, and technological capacity.
Targeted supplementation	Oral magnesium (tablets, powders, fortified products) for vulnerable or deficient groups.	Older adults, pregnant women, individuals with chronic disease or inadequate dietary habits.	Rapid correction of deficiency; easily implemented.	Risk of excessive intake without supervision; bioavailability varies by compound.
Health surveillance	Integration of magnesium assessment (serum or ionized) into national nutrition monitoring and clinical screening.	High-risk populations, hospitalized patients.	Enables early detection and evidence-based interventions.	Requires laboratory infrastructure and additional resources.
Personalized supplementation	Tailoring magnesium intake based on dietary intake, genetics, and comorbidities.	Individuals with metabolic or renal disorders, or genetically predisposed.	Higher efficacy and fewer side effects; supports precision nutrition.	High cost; dependent on advanced diagnostics and clinical expertise.

## Data Availability

No new data were created or analyzed in this study. Data sharing is not applicable to this article.

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
