# Peer review of "Magnesium: Health Effects, Deficiency Burden, and Future Public Health Directions"

_nutrients, 2025, doi:10.3390/nu17223626_

Round 1
Reviewer 1 Report
Comments and Suggestions for Authors
Very good overview article.
I consider the manuscript acceptable as it is.
Evaluation of manuscript ID: nutrients-3954444: Details
The aim of the paper (stated in Introduction) is to give an extensive updated overview on magnesium (Mg) metabolism and health effects of deficiencies and the Mg status worldwide and strategies to improve global burden of deficiencies.
Although previous reviews have discussed this issue, a broader review is justified.
I consider the topic original / relevant to the field, as it does address a specific gap.
It gives a comprehensive review on the topic.
Some condensation of the manuscript could be recommended. (see below).
The conclusions are consistent with the evidence and arguments presented and do address the main question posed.
I have no additional comments on the tables and figures.
As for the title of the manuscript: “--in the 21th century-- “ could be removed.
Part 2. Metabolism is acceptable
Part 3. Role in Physical and Mental Well-Being
Some condensation can be recommended:
3.3 Bone … and 3.6 Dental health (can be merged)
3.4 –Neurological.. and 3.5 Mental .. health (might also be merged)
3.8 Cancer – It could be stated more clearly that this section is most relevant for colo-rectal cancer.
Part 4. Magnesium Status World-Wide -: It could be stated more clearly that we lack good biomarkers
Parts 7 and 8. The headings of these sub-chapters could be simplified
Author Response
Authors' Response to Reviewer 1 Comments
We sincerely thank the reviewer for the careful evaluation of our manuscript and for the constructive comments provided. We highly value these suggestions, as they further improved the clarity and precision of the manuscript. Below, we provide a point-by-point response to each comment. All modifications in the revised version are clearly indicated in Track Changes.
We hope that the reviewer will find the revised version of the manuscript suitable for publication in the journal Nutrients.
Reviewer: Very good overview article. I consider the manuscript acceptable as it is. The aim of the paper (stated in Introduction) is to give an extensive updated overview on magnesium (Mg) metabolism and health effects of deficiencies and the Mg status worldwide and strategies to improve global burden of deficiencies. Although previous reviews have discussed this issue, a broader review is justified. I consider the topic original / relevant to the field, as it does address a specific gap. It gives a comprehensive review on the topic. Some condensation of the manuscript could be recommended. (see below).
The conclusions are consistent with the evidence and arguments presented and do address the main question posed. I have no additional comments on the tables and figures.
Authors: We sincerely thank the reviewer for the positive assessment of our manuscript and for acknowledging the relevance, originality, and comprehensive nature of our review. We also appreciate the suggestion regarding potential condensation of the manuscript. In the revised version, we have carefully revisited the text and have streamlined several sections to enhance conciseness and readability, while preserving scientific accuracy and completeness. We are pleased that the tables and figures were found appropriate and clear. We hope that the revisions made further strengthen the quality and clarity of the manuscript.
Reviewer: As for the title of the manuscript: “--in the 21th century-- “ could be removed.
Authors: We thank the reviewer for this recommendation. According to your suggestion, we have removed the wording “in the 21st century” from the manuscript title.
Reviewer: Part 2. Metabolism is acceptable.
Authors: We thank the reviewer for the positive evaluation of Section 2 (Magnesium Metabolism and Homeostasis).
Reviewer: Part 3. Role in Physical and Mental Well-Being
Some condensation can be recommended:
3.3 Bone … and 3.6 Dental health (can be merged)
Authors: We thank the reviewer for this helpful suggestion. In accordance with your recommendation, we have merged Subsections 3.3 (Bone Health) and 3.6 (Dental Health) into a single consolidated subsection to avoid redundancy and improve conciseness and flow within this part of the manuscript. The newly integrated subsection now appears as Section 3.3 (Bone and Dental Health) in the revised version (lines 215 - 282).
Reviewer: 3.4 –Neurological.. and 3.5 Mental .. health (might also be merged)
Authors: We thank the reviewer for this valuable suggestion. We agree that Sections 3.4 (Migraines and Neurological Health) and 3.5 (Mental Health and Sleep Regulation) address closely related domains and that merging them improves structure and reduces redundancy. Accordingly, we have combined these two subsections into a single integrated subsection in the revised manuscript. The new unified subsection now appears as Section 3.4 (Neurological and Mental Health) (lines 284 - 359).
Reviewer: 3.8 Cancer – It could be stated more clearly that this section is most relevant for colo-rectal cancer.
Authors: We thank the reviewer for this comment. We agree that this clarification is important. In the revised manuscript, we now clearly state at the beginning of this subsection that the evidence is most relevant to colorectal cancer. We have also adjusted the wording throughout the subsection to ensure this emphasis is clear to the reader. The updated subsection is presented in the revised manuscript at lines 451 – 504.
Reviewer: Part 4. Magnesium Status World-Wide -: It could be stated more clearly that we lack good biomarkers
Authors: We thank the reviewer for this comment. We agree that this point needed to be strengthened and stated with greater clarity. We have now revised subsection 4.1. Biomarkers of Magnesium Status to explicitly emphasize that there remains no singular gold standard biomarker for assessing magnesium status globally. The updated subsection is presented in the revised manuscript at lines 571 – 618.
Reviewer: Parts 7 and 8. The headings of these sub-chapters could be simplified
Authors: We thank the reviewer for this suggestion. We agree that the headings in Sections 7 and 8 were unnecessarily long and could be simplified for improved readability and clarity. We have now revised them accordingly.
Section 7 is now titled: “The Central Role of Magnesium in Nutrient Metabolism”
Section 8 is now titled: “Magnesium Deficiency: A Public Health Challenge”

Reviewer 2 Report
Comments and Suggestions for Authors
This article is an addition to the numerous reviews, exemplified by the first 25 references, about the nutritional importance and health benefits of magnesium. Because it will be one among many, it will reduce its impact in the scientific literature. The authors may want to consider the following suggestions and comments as items that might strengthen its content.
Line 56 - The word "despite" is not proper for this sentence. The sentence probably should read "Because of its broad physiological significance magnesium deficiency apparently is highly prevalent."
Line 65 - What is meant by "patterns of prevalence"?
Lines 85-86 - These are just examples of substances that can affect absorption there more - see your reference #1.
Lines 101-110 - This information does not appear appropriate for this section. It is more of an introductory or a conclusion statement than information about magnesium metabolism. It probably should be a separate section, but it is almost redundant to the material that follows.
Line 111 - A better word than "physical" would be "physiological".
Section 3.1 - The discussion about cardiovascular health is too limited. It does not discuss hearth rhythm changes, effect of chronic inflammation, etc. See reference #1.
Lines 165-168 - Do you have a suggestion for the negative results? Could the insulin supplementation have been a factor?
Section 3.6 - This is a good inclusion in the review. Many other reviews do not discuss magnesium in dental health.
Section 3.7 - Perhaps the association of chronic inflammation with chronic disease such as cardiovascular disease could be briefly discussed here.
Section 3.9 - The first sentence in this section has very little relation to the following content. It should be eliminated.
Lines 418-421 - Please give a reference for this statement. See following comment.
Section 4.1 - The authors might make this section more comprehensive by considering the following article: Biological Trace Element Research 177:43-52 and Nielsen, Forrest; Principals of Nutritional Assessment: Magnesium. https://nutritionalassessment.org,magnesium/
Lines 432-434 - Probably should indicate that these are United States and Canada RDAs.
Section 4.2 - It may be appropriate to discuss the findings in American Journal of Clinical Nutrition 2006:84:843-5=852 in this section.
Lines 466-467 - Weight probably should be included based on the article above.
Line 479 -Considering the recent publications about what is a low serum magnesium perhaps it should be stated that the hypomagnesemia statements do not consider the possibility that individuals with serum values of 0.85 mmol/L could be deficient.
Line 537 - What is meant by "4-5%"? The absorption rate for these forms is much higher than that. Moreover, these forms are effective, but may be less efficient than organic forms, for systemic repletion.
Table 3 - It should be noted that the adverse effects with inorganic forms are noted mainly when pharmacological doses are given for the purpose of alleviating constipation. Reasonable doses are absorbed sufficiently to provide additional nutritional magnesium without adverse effects.
Line 551 - I do not understand the basis for the statement "short term use."
Line 627 - I think too much emphasis is put on the use of ionized magnesium as a status indicator. This measure has many of the shortcomings of total serum magnesium. See the second reference listed in the comments for Section 4.1 above.
Line 638 - I believe cardiovascular disease should be added to this list.
Author Response
Authors' Response to Reviewer 2 Comments
We sincerely thank the reviewer for the thoughtful assessment of our manuscript and the valuable suggestions provided. We greatly appreciate the time invested in reviewing our work. The comments raised have contributed to substantial improvements in clarity, conceptual framing, and scientific precision throughout the revised version. Below, we provide a detailed, point-by-point reply to each comment. All revisions made in the manuscript are clearly highlighted in Track Changes.
We hope that the revised version will satisfactorily address all concerns raised and that it will now be considered suitable for publication in Nutrients.
Reviewer: This article is an addition to the numerous reviews, exemplified by the first 25 references, about the nutritional importance and health benefits of magnesium. Because it will be one among many, it will reduce its impact in the scientific literature. The authors may want to consider the following suggestions and comments as items that might strengthen its content.
Authors: We thank the reviewer for this comment. We acknowledge the large number of existing reviews on magnesium and appreciate the suggestions to strengthen the distinct value of this manuscript. We will revise accordingly to better highlight the specific contribution and ensure the content is focused, concise, and adds meaningful perspective to the current literature.
Reviewer: Line 56 - The word "despite" is not proper for this sentence. The sentence probably should read "Because of its broad physiological significance magnesium deficiency apparently is highly prevalent."
Authors: We thank the reviewer for this suggestion. We have revised the sentence accordingly. The revised sentence now reads as follows:
lines 56 – 58: “Because of its broad physiological significance, magnesium deficiency apparently is highly prevalent, largely due to modern dietary patterns, chronic stress, medication use, and certain health conditions [8].”
Reviewer: Line 65 - What is meant by "patterns of prevalence"?
Authors: We thank the reviewer for pointing this out. We agree that this wording was unclear. We have revised the sentence to make it more precise. The revised sentence now reads as follows:
lines 63 – 66: “This review aims to provide a comprehensive synthesis of current knowledge on magnesium metabolism and homeostasis, emphasizing its systemic health effects, the consequences of deficiency, and the global prevalence of low magnesium status.”
Reviewer: Lines 85-86 - These are just examples of substances that can affect absorption there more - see your reference #1.
Authors: We thank the reviewer for this comment. We agree and have now expanded this sentence to include additional examples of dietary components that modulate magnesium absorption as suggested. The sentence has been revised and now reads as follows:
lines 84 – 89: “Intestinal uptake occurs via passive paracellular diffusion and active transcellular transport, and is modulated by dietary composition: phytates, oxalates, high-fiber foods, and high doses of calcium, phosphorus, and iron can inhibit absorption, whereas lactose, certain carbohydrates, like oligosaccharides and inulin, proteins, and medium-chain triglycerides can enhance it [1,19-22].”
Reviewer: Lines 101-110 - This information does not appear appropriate for this section. It is more of an introductory or a conclusion statement than information about magnesium metabolism. It probably should be a separate section, but it is almost redundant to the material that follows.
Authors: We thank the reviewer for this comment. We agree that this information was not optimally placed within this section and overlapped with content addressed later in the manuscript. We have deleted this portion as suggested to avoid redundancy and improve structural coherence.
Reviewer: Line 111 - A better word than "physical" would be "physiological".
Authors: We thank the reviewer for this suggestion. We agree with the proposed wording change and have replaced “physical” with “physiological” in the revised manuscript.
Reviewer: Section 3.1 - The discussion about cardiovascular health is too limited. It does not discuss hearth rhythm changes, effect of chronic inflammation, etc. See reference #1.
Authors: We thank the reviewer for this comment. We agree that the section required additional depth. We have now expanded Section 3.1 Blood Pressure and Cardiovascular Health to include discussion on cardiac rhythm alterations and have added supporting evidence to address this aspect more comprehensively. The impact of chronic inflammation on cardiovascular health is addressed in greater detail in a Section 3.5 Chronic Inflammation to avoid redundancy. Please find the revised version of Section 3.1 in lines 118 – 163.
Reviewer: Lines 165-168 - Do you have a suggestion for the negative results? Could the insulin supplementation have been a factor?
Authors: We thank the reviewer for this remark. We have now added a brief explanation in the revised text that now reads as follows:
lines 194 – 201: “The lack of glycaemic improvement in the latter study may be attributable to the confounding effects of insulin therapy. Exogenous insulin administration can modify insulin sensitivity and β-cell responsiveness, potentially masking the metabolic benefits of magnesium supplementation [51-53]. Magnesium’s role in enhancing insulin sensitivity appears more pronounced in non-insulin-treated individuals, suggesting that treatment context critically influences outcomes [54]. Moreover, insulin therapy itself may contribute to progressive β-cell dysfunction over time, complicating the interpretation of adjunctive interventions such as magnesium supplementation [52].“
Reviewer: Section 3.6 - This is a good inclusion in the review. Many other reviews do not discuss magnesium in dental health.
Authors: We thank the reviewer for this positive remark. We appreciate the acknowledgment and agree that magnesium’s role in dental health is often underrepresented in the literature. We are pleased that this section adds value and unique perspective to the review.
Reviewer: Section 3.7 - Perhaps the association of chronic inflammation with chronic disease such as cardiovascular disease could be briefly discussed here.
Authors: We thank the reviewer for this suggestion. We have now revised Section 3.5 Chronic Inflammation to briefly address the association between chronic inflammation, low magnesium status, and cardiovascular disease risk, as recommended. Please see the revised chapter at lines 414 – 450.
Reviewer: Section 3.9 - The first sentence in this section has very little relation to the following content. It should be eliminated.
Authors: We thank the reviewer for this suggestion. We agree and have removed the first sentence.
Reviewer: Lines 418-421 - Please give a reference for this statement. See following comment.
Authors: We thank the reviewer for this comment. A reference has now been added to support this statement.
Reviewer: Section 4.1 - The authors might make this section more comprehensive by considering the following article: Biological Trace Element Research 177:43-52 and Nielsen, Forrest; Principals of Nutritional Assessment: Magnesium. https://nutritionalassessment.org,magnesium/
Authors: We thank the reviewer for this suggestion. We have now incorporated information from both proposed references into Section 4.1 to make it more comprehensive. Please see the revised Section 4.1 Biomarkers of Magnesium Status at lines 571 – 618.
Reviewer: Lines 432-434 - Probably should indicate that these are United States and Canada RDAs.
Authors: We thank the reviewer for this remark. We have now clarified that the RDA values refer to the United States and Canada. Please see the revised sentence at lines 623 – 626.
Reviewer: Section 4.2 - It may be appropriate to discuss the findings in American Journal of Clinical Nutrition 2006:84:843-5=852 in this section.
Authors: We thank the reviewer for this suggestion. We have now incorporated discussion of findings from Hunt & Johnson to the Section 4.2 (lines 628 – 637).
Reviewer: Lines 466-467 - Weight probably should be included based on the article above.
Authors: We thank the reviewer for this observation. We have now added this aspect to Section 4.2 to reflect that body size may modestly influence magnesium requirements (lines 669 – 672).
Reviewer: Line 479 -Considering the recent publications about what is a low serum magnesium perhaps it should be stated that the hypomagnesemia statements do not consider the possibility that individuals with serum values of 0.85 mmol/L could be deficient.
Authors: We thank the reviewer for this important point. We have now added a clarifying statement in this section indicating that hypomagnesemia prevalence estimates are based on conventional cut-offs and do not capture individuals with serum magnesium in the low-normal range (e.g., <0.85 mmol/L) who may still be deficient. The newly inserted text can be found at lines 686 – 689.
Reviewer: Line 537 - What is meant by "4-5%"? The absorption rate for these forms is much higher than that. Moreover, these forms are effective, but may be less efficient than organic forms, for systemic repletion.
Authors: We thank the reviewer for this observation. We have removed this part from the manuscript.
Reviewer: Table 3 - It should be noted that the adverse effects with inorganic forms are noted mainly when pharmacological doses are given for the purpose of alleviating constipation. Reasonable doses are absorbed sufficiently to provide additional nutritional magnesium without adverse effects.
Authors: We thank the reviewer for this clarification. We have now revised both the text and Table 3 to indicate that adverse effects with inorganic forms mainly occur at pharmacological doses, while reasonable nutritional doses can still contribute to systemic magnesium intake. This part of the manuscript now reads as follows:
lines 761 – 768: “Conversely, inorganic salts such as magnesium oxide, hydroxide, and sulfate are less bioavailable than most organic forms, but adverse gastrointestinal effects are mainly seen when these forms are used at pharmacological doses for the purpose of alleviating constipation. At reasonable doses, they can be absorbed sufficiently to contribute to magnesium intake without necessarily producing laxative effects [193,200-202]. These forms may therefore be suitable for gastrointestinal use but may be less efficient for systemic repletion.”
|
Magnesium form |
Type |
Bioavailability |
Common applications |
Typical adverse effects |
References |
|
Magnesium citrate |
Organic |
High |
General supplementation, fatigue, constipation, muscle cramps |
Loose stools at high doses |
[193,194] |
|
Magnesium glycinate (bisglycinate) |
Organic |
Very high |
Stress reduction, sleep, muscle relaxation, sensitive stomach |
Generally well tolerated |
[92,196] |
|
Magnesium malate |
Organic |
High |
Fatigue, muscle pain, athletic recovery |
Generally well tolerated |
[202,203] |
|
Magnesium taurate |
Organic |
High |
Cardiovascular and metabolic support |
Generally well tolerated |
[33] |
|
Magnesium orotate |
Organic |
High |
Heart health, energy metabolism |
Expensive, generally well tolerated |
[92] |
|
Magnesium threonate |
Organic |
High |
Cognitive and neurological support |
Gastrointestinal discomfort, rarely lethargy |
[198,199,202,204,205] |
|
Magnesium ascorbate |
Organic |
High |
Combined magnesium and vitamin C supplementation, immune support |
Mild gastrointestinal discomfort |
[206] |
|
Magnesium lactate |
Organic |
Moderate-high |
General supplementation |
Mild gastrointestinal discomfort |
[207] |
|
Magnesium aspartate |
Organic |
High |
Metabolic and electrolyte balance |
Mild gastrointestinal discomfort |
[207] |
|
Magnesium oxide |
Inorganic |
Low |
Pharmacological use for constipation; may contribute nutritionally at reasonable doses |
Loose stools, gas mainly at higher pharmacological doses |
[194] |
|
Magnesium hydroxide |
Inorganic |
Low |
Pharmacological laxative use |
Loose stools mainly at higher pharmacological doses |
[200] |
|
Magnesium sulfate |
Inorganic |
low (oral); high (intravenous) |
Intravenous therapy, baths for muscle relaxation, oral use primarily pharmacological laxative |
Strong laxative effect orally only at pharmacological doses |
[200,201] |
|
Magnesium chloride |
Inorganic |
Moderate-high |
General supplementation, topical preparations |
Skin irritation (in topical form), gastrointestinal upset possible at higher doses |
[208] |
Reviewer: Line 551 - I do not understand the basis for the statement "short term use."
Authors: We thank the reviewer for this comment. We have now removed the wording “short-term use” and revised the text to clarify that magnesium oxide is mainly used when a laxative effect is desired, rather than implying a limitation in duration of use. The revised sentence now reads as follows:
lines 783 – 786: “Supplement selection should therefore balance bioavailability, safety, and affordability—with citrate and glycinate preferred for efficiency and tolerance, and oxide serving as an accessible, low-cost option particularly when a laxative effect is desired.”
Reviewer: Line 627 - I think too much emphasis is put on the use of ionized magnesium as a status indicator. This measure has many of the shortcomings of total serum magnesium. See the second reference listed in the comments for Section 4.1 above.
Authors: Thank you for this comment. We have revised the sentence to avoid overemphasizing ionized magnesium as a superior indicator and now refer more broadly to the need for refinement and standardization of physiologically meaningful indicators beyond total serum magnesium. The revised sentence now reads as follows:
lines 880 – 884: “While routine serum measurements remain widely used, future progress will require the refinement and standardization of physiologically meaningful indicators beyond tMg, which may enhance diagnostic accuracy in both population monitoring and high-risk clinical settings [230].
Reviewer: Line 638 - I believe cardiovascular disease should be added to this list.
Authors: Thank you for this comment. We have now added cardiovascular disease to the list to reflect its well-established association with low magnesium status.

Reviewer 3 Report
Comments and Suggestions for Authors
The manuscript provides an overview of magnesium physiology and its relevance to human health. The topic is timely and of general interest; however, the current version does not meet the standards expected of a high impact review. The paper is largely descriptive, lacks critical synthesis, and fails to establish a clear conceptual framework or novel insights.
Major Comments
- The manuscript compiles a vast body of literature but provides little critical evaluation or integration. Discussion of underlying mechanisms is minimal, and there is no comparison between studies. Consequently, the paper offers a superficial overview rather than an interpretative or hypothesis-driven narrative.
- Transitions between sections are missing, and the overall logical progression is weak. While the introduction is adequate, it quickly devolves into repetitive summaries. Sections 3 and 4 read as disconnected mini reviews, lacking integration under a coherent central thesis.
- The current formatting obscures key messages. The presentation is redundant. Several sections reiterate similar mechanistic explanations (e.g. ATP stabilization and magnesium’s cofactor role) with minimal differentiation. The review could be substantially condensed without losing essential content.
- Critical discussion is absent. The manuscript does not address limitations of current evidence, inconsistencies among studies, or major research gaps. The conclusion merely restates well established facts instead of proposing a conceptual synthesis or outlining future directions.
Comments on the Quality of English Language
While grammar and syntax are generally correct, the writing is monotonous and overly descriptive. The tone is encyclopedic rather than analytical.
Author Response
Authors' Response to Reviewer 3 Comments
We sincerely thank the reviewer for the thoughtful evaluation of our manuscript and the constructive comments provided. We highly appreciate the time and effort invested in reviewing our work. The suggestions have contributed to further improving clarity, precision, and scientific robustness. Below, we provide a point-by-point response to each comment. All revisions are indicated in Track Changes within the revised manuscript.
Reviewer: The manuscript provides an overview of magnesium physiology and its relevance to human health. The topic is timely and of general interest; however, the current version does not meet the standards expected of a high impact review. The paper is largely descriptive, lacks critical synthesis, and fails to establish a clear conceptual framework or novel insights.
Authors: We thank the reviewer for this overall assessment and appreciate the constructive feedback. We have carefully revised the manuscript with the aim of improving clarity, focus and interpretative value, and have introduced several modifications throughout the text to strengthen synthesis of the evidence. We hope that the revised version addresses the reviewer’s concerns and improves the overall contribution of the review.
Reviewer: The manuscript compiles a vast body of literature but provides little critical evaluation or integration. Discussion of underlying mechanisms is minimal, and there is no comparison between studies. Consequently, the paper offers a superficial overview rather than an interpretative or hypothesis-driven narrative.
Authors: We thank the reviewer for this comment. We appreciate the need to go beyond description and have revised the manuscript to strengthen interpretative aspects, expand mechanistic context where relevant, and improve integration between study findings. Comparative elements have been added in several sections to better reflect consistencies and discrepancies in the evidence. We hope that these revisions improve the scientific value and interpretative depth of the review.
List of the changes made:
In Section 2. Magnesium Metabolism and Homeostasis, we have now added a brief integrative statement linking intestinal, renal, and skeletal regulation to systemic deficiency potential, in order to enhance mechanistic interpretation and synthesis (lines 103 – 106).
Within Section 3. Magnesium and Its Role in Physiological and Mental Well-Being, we have updated text at multiple locations to improve critical evaluation, synthesis, and conceptual integration (lines 140 – 143; lines 152 – 154; lines 202 – 205; lines 231 – 235; lines 245 – 249; lines 272 – 274; lines 303 – 306; lines 315 – 318; lines 335 – 337; lines 434 - 436; lines 441 – 443; lines 466 – 468; lines 473 – 475; lines 493 – 496; lines 520 – 526; lines 551 – 554).
In the revised version, Section 4.1 Biomarkers of Magnesium Status has been substantially expanded to provide clearer critical interpretation of biomarker limitations and to integrate additional evidence, including functional biomarkers and the implications of emerging cut-off proposals. We clarified where serum magnesium may underestimate true deficiency, included discussion on the role and constraints of ionized magnesium, and strengthened the synthesis of the comparative value of existing measures. We believe that these changes enhance the interpretive depth of the section while maintaining a balanced and evidence-based framing.
In Section 4.2 Dietary Requirements we have incorporated controlled metabolic balance data to contrast physiological requirements versus population reference values and explicitly addressed body size as an additional determinant.
In Section 4.3 Determinants and Global Patterns of Magnesium Deficiency we revised the hypomagnesemia paragraph to add interpretative context, including the relevance of low-normal serum magnesium cut-offs (< 0.85 mmol/L) and mechanisms contributing to higher prevalence in hospitalized patients. We believe these revisions collectively strengthen the analytical perspective and improve integration across subsections.
An interpretative summary paragraph has been added at the end of Section 5. Dietary Sources of Magnesium to integrate dietary composition, food matrix effects, and preparation practices with implications for magnesium bioavailability and population strategies (lines 727 – 732).
We have expanded Section 6.1 Bioavailability of Magnesium Compounds by adding a statement highlighting the clinical implications of formulation-specific bioavailability differences and the need for contextual selection rather than assuming interchangeability. This provides more critical evaluation beyond descriptive listing of individual formulations (lines 768 – 772).
In Section 8. Magnesium Deficiency: A Public Health Challenge we expanded the public health synthesis by more explicitly linking magnesium deficiency to structural dietary and systems-level determinants, and emphasized that persistent insufficiency reflects broader nutrition transitions rather than purely individual intake (lines 871 – 874). We also strengthened the forward-looking perspective by clarifying the relevance of biomarker standardization to population surveillance and health policy (lines 880 – 884).
Reviewer: Transitions between sections are missing, and the overall logical progression is weak. While the introduction is adequate, it quickly devolves into repetitive summaries. Sections 3 and 4 read as disconnected mini reviews, lacking integration under a coherent central thesis.
Authors: We thank the reviewer for this comment. We have now added integrative transition sentences at key section endings to improve narrative flow and better connect physiological, clinical and public health themes across the manuscript. We believe these additions strengthen coherence between topics as requested. Those transitions were added at the following locations in the revised version of the manuscript:
lines 561 – 568: transition from Section 3 to Section 4.
lines 703 – 707: transition from Section 4 to Section 5.
lines 733 – 737: transition from Section 5 to Section 6.
lines 787 – 794: transition from Section 6 to Section 7.
lines 850 – 856: transition from Section 7 to Section 8.
Reviewer: The current formatting obscures key messages. The presentation is redundant. Several sections reiterate similar mechanistic explanations (e.g. ATP stabilization and magnesium’s cofactor role) with minimal differentiation. The review could be substantially condensed without losing essential content.
Authors: We thank the reviewer for this important comment. We agree that mechanistic repetition reduced clarity, particularly with recurrent references to ATP stabilization and magnesium’s cofactor role. We have now removed or shortened several redundant mechanistic sentences (Section 2 lines 73 - 77, Section 3.1 lines 121 - 123, Section 3.8 lines 535 – 536, and Section 7 lines 798 - 800) to improve focus and streamline key messages. Also, in line with the request of Reviewer 1 and to enhance coherence, we merged the subsections on bone & dental health and neurological & mental health within Section 3 to reduce fragmentation and improve integration. We believe these changes have substantially improved readability, reduced overlap, and strengthened conceptual flow throughout the manuscript.
Reviewer: Critical discussion is absent. The manuscript does not address limitations of current evidence, inconsistencies among studies, or major research gaps. The conclusion merely restates well established facts instead of proposing a conceptual synthesis or outlining future directions.
Authors: We thank the reviewer for this comment. We have now strengthened critical evaluation across all major sections. Specifically, we have incorporated discussion of study heterogeneity, methodological limitations, inconsistent RCT findings, context-dependency of effects, and confounding factors throughout Sections 3 and 4. We believe these additions provide clearer interpretation of the evidence base and address the limitation that the prior version was overly descriptive. Regarding the Conclusions, we agree that the original conclusion section was overly descriptive and did not sufficiently synthesize limitations or articulate forward-looking directions. In the revised manuscript, the Conclusions section has been substantially rewritten to provide a more interpretive and integrative synthesis. The new version (lines 892 – 915) now explicitly discusses key limitations of the current evidence base, highlights major research gaps (including biomarker standardization and phenotype-specific responses to supplementation), and outlines priority directions for future investigation.
Reviewer: The English could be improved to more clearly express the research.
Authors: We thank the reviewer for this helpful comment. We have carefully revised the manuscript throughout to improve clarity and precision of the English. Grammar, readability, and structural flow have been edited in all sections to ensure the intended research message is expressed more clearly. We believe the revised version is substantially improved as a result of these changes.

Round 2
Reviewer 2 Report
Comments and Suggestions for Authors
This is a much-improved manuscript. It now serves as a good review of the current status of magnesium.
Reviewer 3 Report
Comments and Suggestions for Authors
The revised manuscript represents a notable improvement over the previous submission. The structure is now clearer, the logical flow between sections is more coherent, and the scientific language has been refined considerably. Citations have been updated and standardized, and the addition of mechanistic explanations and recent meta-analyses enhances the manuscript’s scientific depth.
The work now presents a more comprehensive and balanced overview of magnesium metabolism and its physiological and clinical implications. However, despite this progress, several issues remain that must be addressed before the paper can be considered for publication.
1- Consider providing a synthetic graphical abstract or summary figure outlining the main biological mechanisms and clinical outcomes linked to magnesium.
2- The writing is significantly improved, but certain paragraphs remain overloaded with citations and minor redundancies (e.g., repeated mentions of “magnesium as a cofactor in >300 enzymes”). These should be streamlined for clarity.
3- While the authors summarize a vast amount of data, critical synthesis remains limited. In several subsections (e.g., diabetes, hypertension, mental health), the evidence is presented descriptively without sufficient discussion of study quality, confounding factors, or limitations of meta-analyses.
4- Please ensure uniform reference formatting (some [xx] brackets appear with inconsistent spacing or line breaks).
5- Supplementation. This part is much improved and now provides a well-balanced overview of magnesium compounds. However, it could benefit from a brief critical note on safety thresholds, upper intake limits, and the risk of hypermagnesemia in renal impairment.
Comments on the Quality of English Language
English is clear and technically accurate, though some minor grammatical corrections are still needed (“By contrast” and “Conversely” are sometimes overused).
